# Mutable Collagenous Tissue: A Concept Generator for Biomimetic Materials and Devices

**DOI:** 10.3390/md22010037

**Published:** 2024-01-07

**Authors:** M. Daniela Candia Carnevali, Michela Sugni, Francesco Bonasoro, Iain C. Wilkie

**Affiliations:** 1Department of Environmental Science and Policy, University of Milan, 20133 Milan, Italy; daniela.candia@unimi.it (M.D.C.C.); michela.sugni@unimi.it (M.S.); francesco.bonasoro@unimi.it (F.B.); 2School of Biodiversity, One Health and Veterinary Medicine, University of Glasgow, Glasgow G12 8QQ, UK

**Keywords:** biomimetic nanocomposites, collagen, juxtaligamental cell, mechanically tunable implants, proteoglycan, soft actuators, soft robotics, stimuli-responsive material, tensilin

## Abstract

Echinoderms (starfish, sea-urchins and their close relations) possess a unique type of collagenous tissue that is innervated by the motor nervous system and whose mechanical properties, such as tensile strength and elastic stiffness, can be altered in a time frame of seconds. Intensive research on echinoderm ‘mutable collagenous tissue’ (MCT) began over 50 years ago, and over 20 years ago, MCT first inspired a biomimetic design. MCT, and sea-cucumber dermis in particular, is now a major source of ideas for the development of new mechanically adaptable materials and devices with applications in diverse areas including biomedical science, chemical engineering and robotics. In this review, after an up-to-date account of present knowledge of the structural, physiological and molecular adaptations of MCT and the mechanisms responsible for its variable tensile properties, we focus on MCT as a concept generator surveying biomimetic systems inspired by MCT biology, showing that these include both bio-derived developments (same function, analogous operating principles) and technology-derived developments (same function, different operating principles), and suggest a strategy for the further exploitation of this promising biological resource.

## 1. Introduction

Collagenous tissue is an important structural biomaterial in most multicellular animals. The mechanical properties of collagenous tissue are usually relatively stable, with any significant non-pathological changes taking place over timescales of days (during relaxation of the mammalian uterine cervix [1]) to years (during maturation and ageing [2]). However, all five living classes of the phylum Echinodermata (sea lilies and featherstars, sea-urchins, sea-cucumbers, starfish, brittlestars) possess collagenous tissue that can drastically alter its mechanical properties within a timescale of seconds under the direct control of the nervous system. Although such mutable collagenous tissue (MCT) appears to be unique to echinoderms, some specific features underpinning its mechanical adaptability—particularly the absence of permanently stable crosslinks between its constituent collagen fibrils—may be ancestral and may have permitted the emergence of similar phenomena in other phyla. There are, for example, parallels between MCT and the collagenous mesohyl of demosponges whose tensile properties are under non-neural physiological control [3].

MCT demonstrates a micro-architectural diversity comparable to that of vertebrate fibrous connective tissue, occurring as three-dimensional fiber networks in dermal layers, parallel-fibered ligaments linking skeletal components, and crossed-fiber helical arrays in the walls of tubular organs. These anatomical structures perform the same functions as their vertebrate equivalents: they resist, transmit and dissipate mechanical forces, and they store and release elastic strain energy. Their variable tensility, however, adds another dimension to their functional versatility, which is of widespread importance to echinoderm biology and may have contributed to the evolutionary success of the phylum. In echinoderms, prolonged postural fixation depends on passive MCT stiffening rather than active muscle contraction, resulting in a considerable energy saving [4,5]. Another process associated with MCT is irreversible destabilization, which enables defensive self-detachment (autotomy) [6] and asexual reproduction by division of the whole body (fission) in brittlestars, starfish and sea-cucumbers [7,8,9]. Possibly related is the ‘autolysis’ or ‘melting’ of the whole dermis exhibited by some sea-cucumbers in adverse conditions, a pathological phenomenon of great commercial importance [10,11,12].

Intensive research on MCT began over 50 years ago [13,14,15] and over 20 years ago, MCT first inspired a biomimetic design [16]. Since then, MCT, and sea-cucumber dermis in particular, has inspired the development of many artificial materials and devices (see, e.g., [17,18,19]).

This review was compiled primarily for the benefit of bioscientists who are unfamiliar with MCT biology and its relevance for recent biomimetic research and development. We first provide an up-to-date account of present knowledge of MCT, including the molecular mechanisms regulating its variable tensility. We then survey biomimetic solutions that have been derived from MCT-related structural and functional concepts, and suggest a strategy to inspire further design.

## 2. Structural Organization of MCT

### 2.1. Extracellular Components

The MCT of all echinoderms consists predominantly of extracellular materials and includes a relatively small volume fraction of cellular components (Figure 1A). With one exception, the extracellular materials comprise mainly transversely banded collagen fibrils aggregated into bundles (fibers) accompanied by loose arrangements of microfibrils and interfibrillar proteoglycans (Figure 1B,C). The exception is the tendon tissue of the intervertebral muscles at the autotomy planes of brittlestar arms, which is an extension of the basement membrane of the muscle cells (see [20] for further information).

The banded collagen fibrils of MCT, like those of vertebrate connective tissue, are parallel arrays of trimeric collagen molecules with a regular stagger between adjacent molecules ranging from 40 to 80 nm, a much wider variability than the 65–67 nm reported for vertebrate fibrils [23,24]. The collagen molecules of most echinoderm fibrils comprise two fibrillar α chains (1α and 2α) which form (1α)_2_2α heterotrimers [25,26,27]. A small proportion of collagen molecules in the fibrils of sea-urchin MCT contain a third fibrillar chain (5α) and have a (1α)_2_5α stoichiometry [25]. The 5α chain is unusual in that its N-propeptide is not removed prior to fibril assembly, as occurs in the echinoderm 2α chain and in all vertebrate fibrillar procollagens [28], and is located at the surface of the fibrils. The 5α N-propeptide is also notable because it contains 11 SURF (‘sea-urchin fibrillar’) modules, which are also present in the 2α N-propeptide and in fibrosurfin—an interfibrillar protein of unknown function. Since cleaved 2α N-propeptides have been immunolocalized to the periphery of fibril bundles in a sea-urchin mutable ligament [29], it has been suggested that these SURF-containing molecules play a role in MCT variable tensility [25].

So far, there has been only a very limited application of comparative ‘-omics’ methodologies to the elucidation of MCT molecular organization. Surveys of the *Strongylocentrotus purpuratus* genome and transcriptome analysis of a sea-cucumber body wall revealed no unusual features of the extracellular matrix (ECM) that could explain the mechanical adaptability of MCT. For example, echinoderms have up to four fibrillar collagen genes of the vertebrate I/II/III type and two of the V type, all of which encode molecules occurring in varying combinations in the banded fibrils of vertebrates [9,30,31]. These investigations also demonstrated the presence of fibrillin genes and their transcripts, complementing biochemical and immunological evidence that the beaded microfibrils that are ubiquitous in MCT and non-mutable echinoderm ligaments [22,24,32] consist at least partly of fibrillin-like proteins. These microfibrils may facilitate slippage between adjacent fibril bundles during MCT deformation and contribute to passive elastic recoil after the removal of external forces [33].

Other interfibrillar components include molecules that are responsible for the cohesion between adjacent collagen fibrils and therefore have a major influence on the mechanical properties of MCT. Proteoglycans, which consist of a protein core and glycosaminoglycan (GAG) sidechains, are present both within and on the surface of the collagen fibrils [33] (Figure 1C). There is biochemical evidence that surface proteoglycans act as binding sites for other molecules that form interfibrillar crossbridges. Staining with the cationic dyes cuprolinic blue or cupromeronic blue, which label GAG sidechains, reveals both punctate/globular precipitates on the surface of fibrils and linear structures that extend between adjacent fibrils and are attached to specific sites within each fibril D-period [24,34,35]. In featherstar ligaments and sea-cucumber dermis, the GAG components of these surface proteoglycans have been identified as chondroitin sulfate [34,35]. MCT contains several other molecules that contribute to interfibrillar cohesion but whose extracellular disposition is unknown; these are discussed below.

### 2.2. Cellular Components

The most abundant and characteristic cellular components of MCT are neurosecretory-like cell processes containing large (diameter > ca. 100 nm) dense-core vesicles (LDCVs) (Figure 1A,D). These were first described in the brittlestar intervertebral ligament and named ‘juxtaligamental cells’ (JLCs), because in brittlestars, the perikarya are always located outside, but usually closely adjacent to, collagenous structures (Figure 1E) [36]. In almost all investigated mutable collagenous structures, there are two or more populations of LDCV-containing processes distinguished by the size, and sometimes the shape and electron density, of their LDCVs (Figure 1D) [22,37,38,39,40].

An expanding body of evidence indicates that JLCs are neurons. In brittlestars, they are located in aggregations, known as juxtaligamental nodes, that have a central, neuropil-like region penetrated by the axons of motoneurons and chemical synapses between axonal and juxtaligamental processes (Figure 1E–H) [22,37,41,42,43]. Juxtaligamental nodes have an outer capsule of neuroglia-like cells with centrally directed partitions that compartmentalize the juxtaligamental perikarya (Figure 1F). This suggests that the nodes could be ganglionic integrating centers that coordinate changes in the tensile properties of MCT with the activities of other effector systems [22]. Less organized cellular aggregations with ganglion-like features are associated with MCT in other echinoderm classes [44].

It is highly likely that JLCs are the effectors that directly alter the tensile properties of MCT, since their processes terminate in MCT, have no possible cellular targets, and link the ECM to the motor nervous system, and since they are also absent from non-mutable echinoderm collagenous structures [33]. Putative effector molecules occur in sea-cucumber JLCs. The stiffening proteins tensilin and stiparin have been immunolocalized to the LDCVs of JLCs in the dermis [16,45] (also Keene and Trotter, unpublished data), and Demeuldre et al. [40] detected tensilin by immunohistochemistry and in situ hybridization in JLCs in the connective tissue of the Cuvierian tubules (extrudable adhesive structures used for defense [46]). In some mutable collagenous structures, alterations in tensile state are accompanied by changes in JLC ultrastructure. These usually include indications that LDCVs or their contents are released into the extracellular compartment. Such changes have been seen mainly in structures undergoing an irreversible alteration in mechanical properties, which can be in the form of either drastic weakening (as occurs during autotomy) or stiffening (as undergone by Cuvierian tubules after expulsion) [24,37,40,47].

Other cellular components of MCT include heterogeneous vacuole-containing cells (Figure 1A), which may be phagocytic, and, in a few starfish and sea-urchin structures, myocytes [33,48]. Most echinoderm collagenous tissue, including MCT, appears to lack fibroblasts [22,39,40,47,49]. As echinoderms show indeterminate growth [50,51], they must possess as yet unidentified populations of cells with the capacity to maintain the continuous expansion of connective tissue structures.

## 3. Mechanisms of Tensile Change

Mutable collagenous structures show four patterns of tensile change: (1) only irreversible destabilization (as occurs during autotomy, e.g., crinoid syzygial ligament), (2) irreversible destabilization, as well as reversible stiffening and destiffening (e.g., ophiuroid intervertebral ligament), (3) only reversible stiffening and destiffening (e.g., crinoid brachial ligament), and (4) only irreversible stiffening (e.g., holothurian Cuvierian tubules) [6,40]. These varying capacities are made possible by an important feature of MCT that distinguishes it from the collagenous tissues of vertebrates: interfibrillar crosslinking in the latter is highly dependent on covalent molecular interactions and is relatively stable, whereas in MCT, interfibrillar crosslinking is labile and under physiological control. This is illustrated by the contrast between the extractability of echinoderm collagen fibrils, which can be isolated by mild chemical and mechanical methods [52,53], and the inextractability of intact collagen fibrils from normal adult vertebrate collagenous tissues (although molecular collagen can be isolated easily from the latter) [54,55,56].

In recent years, most ideas on the possible molecular mechanisms underpinning the physiological control of MCT variable tensility have been derived from investigations of sea-cucumber dermis and have employed a model based on the idea that the tissue can adopt three different mechanical states: compliant, standard and stiff. These states show qualitatively different features, which suggests that different mechanisms are involved in shifts between standard and compliant states and shifts between standard and stiff states [57]. Research on sea-cucumber dermis has focused on chemical factors that can be isolated from it and that influence its mechanical behavior in vitro (Figure 2). The best characterized are the tensilins, which have a high degree of sequence identity to TIMP (tissue inhibitor of metalloproteinase) proteins (Figure 3). Tensilins can be isolated only after dermis is subjected to treatments that cause cell lysis, indicating that they are stored intracellularly. They cause calcium-independent aggregation of isolated collagen fibrils and stiffen samples of whole dermis in vitro. By comparing the properties of intact and truncated recombinant tensilin of *Holothuria forskali*, Bonneel et al. [45] recently provided evidence that the N-terminal TIMP-like domain interacts strongly with GAGs attached to the surface of collagen fibrils, and that crossbridging of fibrils may result from the dimerization or oligomerization of the protein mediated by its C-terminal regions (Figure 4).

Sea-cucumber tensilins have a limited stiffening effect; Tamori et al. [58] found that the tensilin from *Holothuria leucospilota* applied in vitro could convert dermis from the compliant to the standard state but not from the standard to the stiff state. The phylum-wide significance of tensilins is also uncertain, since a sea-urchin tensilin has no consistent effect on the mutable compass depressor ligament of the sea-urchin *Paracentrotus lividus* [59]. A second stiffening protein—‘novel stiffening factor’ (NSF)—converts sea-cucumber dermis from the standard to the stiff state but has no effect on compliant dermis and does not aggregate isolated collagen fibrils [60]. The differential stiffening effects of tensilin and NSF support the view that separate molecular mechanisms are responsible for the compliant→standard and standard→stiff transitions. Another protein, stiparin, causes calcium-independent aggregation of collagen fibrils, but has no effect on whole dermis. It is extractable by prolonged immersion of dermis in sea water alone and is the most abundant soluble glycoprotein in the dermis of *Cucumaria frondosa*. Stiparin is thought to be a constitutive component of the ECM that maintains a basal level of interfibrillar cohesion, while allowing slippage between adjacent collagen fibrils [16].

Destiffening molecules have also been extracted from sea-cucumber dermis. ‘Stiparin inhibitor’ is a 62 kDa sulfated glycoprotein that binds stiparin and inhibits its fibril aggregating activity. ‘Plasticiser’ is a small (<15 kDa) protein that has a direct destiffening effect on the dermal ECM and, like tensilin, is released only after cytolytic treatments [16]. Softenin is a ca. 20 kDa destiffening protein that is extractable without cell lysis. As it disaggregates tensilin-aggregated collagen fibrils in vitro and reversibly destiffens cell-dead dermis in the standard state, it may compete for tensilin binding sites on the collagen fibrils [61]. Because of the reversibility and short time course of its effect, softenin is unlikely to be an enzyme. However, enzymes have been suggested as possible MCT destiffeners, particularly matrix metalloproteinases (MMPs), which are extensively involved in the ECM remodeling that accompanies echinoderm development and regeneration [62,63,64]. The synthetic MMP inhibitor galardin was found to stiffen sea-urchin compass depressor ligaments in all three mechanical states, though its effect was much lower on stiff ligaments than on compliant and standard ligaments. Ribeiro et al. [65] speculated that ligament stiffness is adjusted through crosslink degradation by constitutive MMP activity regulated by the cellular release of an endogenous MMP inhibitor. (Figure 5). MMPs have also been implicated in the phenomenon of sea-cucumber dermal liquefaction (‘autolysis’). Although the activity of several endogenous proteinases increases in liquefying dermis [10,11], liquefaction is blocked by MMP inhibitors and only MMP activity achieves the complete disaggregation of collagen fibers into smaller fibril bundles and fibrils (possibly via the degradation of interfibrillar proteoglycans) that characterizes the process [11,12,66,67].

The mechanical behavior of MCT preparations is affected in vitro by several endogenous peptides. Most of their effects are likely to be cell mediated. However, it has been suggested that two peptides, holokinin-1 and holokinin-2, whose heptapeptide sequence is present in the C-terminal domain of 5α collagen, destiffen the dermis by disrupting putative interfibrillar linkages involving 5α collagen [68].

More evidence that separate stiffening mechanisms are responsible for the compliant→standard and standard→stiff transitions has been obtained from comparisons of the water content and distribution in MCTs in different mechanical states. On the basis of measurements of volume and mass, Tamori et al. [69] concluded that water moves out of sea-cucumber dermis during the standard→stiff shift but not during the compliant→standard shift. Similarly, confocal Raman spectroscopy has shown that water moves from the interior to the surface of sea-urchin compass depressor ligaments during the standard→stiff shift but not the compliant→standard shift [70]. Furthermore, in both these mutable collagenous structures the standard→stiff transition alone is accompanied by a significant increase in the collagen fibril packing density [24,71]. The transmission electron microscope observations of Tamori et al. [71] also revealed that in both transitions in sea-cucumber dermis, there is a significant increase in the number of crossbridges connecting adjacent collagen fibrils, leading the authors to propose that dermal stiffening is achieved by three mechanisms: increased crossbridge formation in both transitions, tensilin-dependent stiffening of collagen fibrils (through subfibril fusion) in the compliant→standard transition, and, in the standard→stiff transition, increased bonding within the proteoglycan matrix between the fibril bundles (i.e., fibers), which increases fibril packing density and causes water exudation (Figure 6).

A notable feature of the model of Tamori et al. [71] is the hypothesis that the variable tensility of holothurian dermis involves changes in the stiffness of collagen fibrils resulting from their disaggregation into subfibrils and subsequent reaggregation. Such a mechanism was previously proposed by Erlinger et al. [34] with regard to featherstar ligaments. However, using time-resolved synchrotron small-angle X-ray scattering (SAXS) to compare the behavior of fibrils during in situ tensile loading of holothurian dermis in three mechanical states, Gupta and co-workers [72,73,74] demonstrated that the variable tensility of the tissue can be attributed to modulation of interfibrillar matrix stiffness alone without any contribution from changes in fibril stiffness. This disagreement about the significance of fibril mechanics indicates that, despite holothurian dermis being by far the most thoroughly investigated mutable collagenous structure, there remains considerable uncertainty about the molecular mechanisms responsible for its mechanical adaptability. Furthermore, in view of the relative paucity of information on the MCT of other echinoderm classes [20], the wider applicability of data and concepts generated by research on holothurian dermis (as encapsulated in Figure 2) cannot at present be assessed with any confidence.

## 4. MCT-Inspired Biomimetic Systems

Since 2000, when Trotter et al. [16] first introduced the notion of MCT-inspired synthetic systems, interest in the potential applications (biomedical and others) of MCT has been steadily growing, leading to the opening of an applied multidisciplinary research area continuously expanding according to two substantially different approaches [20]: (1) using extracted MCT components for deployment in synthetic biomedical materials—the biotechnological approach (see, for example, references [75,76,77])—and (2) utilizing MCT as a concept generator—the biomimetic approach, which addresses knowledge transfer from biology to technical applications [78] and is the theme of this review. There is currently a tendency to confuse these two approaches and not to consider them as separate and different fields, the term ‘biomimetic’ being often employed ambiguously in both of these cases.

The biomimetic approach imitates natural models without using their actual constituents, regarding both their structure and functioning as inspiration for solving technical problems and as sources for innovation and sustainable development of human activities and technologies [78,79]. Through the biomimetic approach, the structures, processes and working principles of an organism are identified as concepts (i.e., are concept generators), abstracted into models and eventually transferred to technical applications [80], thus providing models and tools for the innovation process. Living organisms are therefore only indirectly involved in the production of biomimetic products or the development of biomimetic processes. The basic concept of bioinspired design is universally applicable and has been widely employed in applied sciences, from robotics to architecture and computing. Independently of the source of inspiration, it can be applied at any hierarchical bio-level, from nanostructural and microscopical to macrostructural levels. Moreover, and most importantly, besides contributing to the development of new materials and technologies, the biomimetic approach plays an important role in improving knowledge of nature per se through a process of ‘reverse biomimetics’ [78]; in fact, a deeper integrated interdisciplinary analysis provides more consistent and reliable morpho-functional descriptions/understanding of biological systems and their specific adaptations, therein leading to a more comprehensive interpretation of their evolutionary significance. Following advances in knowledge of natural systems, biomimetics now promises immense applied potential, especially in the area of bio-robotics, due to the advanced employment of both powerful software and sophisticated technologies.

In MCT research, as in other applied fields, the biomimetic approach can be adopted for biology-inspired developments (biology-push processes) or technology-derived developments (technology-pull processes) (for a review, see [78]). Although they are both types of biomimetic development in which technical solutions are inspired by living organisms, the two processes differ in the nature of their starting point: this is a biological question in the former case and a technical problem in the latter case.

### 4.1. Biology-Inspired Developments: Biology-Push Processes

Here, the starting point for knowledge transfer is an inspiring idea derived from the structural features and/or operating principles of a specific biological model, which leads to a biomimetic development that benefits bioengineering, clinical practice or healthcare [78,79,81].

With regard to MCT as a concept generator, the most ‘popular’ in a range of potential echinoderm models has been sea-cucumber dermis (see also [17,20]). Biomaterials that mimic extracellular matrix (ECM) can provide a compatible microenvironment to improve efficacy in tissue repair; due to their mechanical adaptability, microstructural interconnectivity and inherent bioactivity, these materials appear to be ideal for the design of living implants for specific applications in tissue engineering (TE) and regenerative medicine.

Trotter et al. [16] first proposed, designed and tested a synthetic analog of sea-cucumber dermis that was composed of collagen fibrils in an artificial elastomeric matrix and included reversible interfibrillar crosslinks formed by photo- or electro-sensitive reagents. Although in its first version this model was a hybrid, since it was inspired by MCT mechanical properties as a concept generator (biomimetic approach) but incorporated natural fibrils extracted from sea-cucumber dermis (biotechnological approach), the final version was completely biomimetic and employed a fully synthetic fibrous composite with dynamically controlled stiffness (Figure 7).

The biology-inspired strategy of Trotter et al. [16] was a source of inspiration for a series of technology-derived developments (see Section 4.2). Wilkie [33] considered potential uses for both biotechnological and biomimetic approaches. In particular, biomimetic applications in the treatment of fibrotic lesions related to joints, tendons and ligaments were hypothesized [81,82,83,84,85] (see Section 4.5). Xia [86], focusing on sea-cucumber dermis, suggested that to mimic the striking stiffening/destiffening performance of MCT, new artificial adaptive materials could be developed by employing a framework of nanofibers immersed in a soft matrix, with reversible bonds between the nanofibers, with stress transfer being controlled by modifying interactions between nanofibers and/or between nanofibers and the matrix polymer via external stimuli. The same concepts were developed by Mo et al. [72], who highlighted again how the adaptive mechanical properties of MCT can provide new practical perspectives for the treatment of pathologies of connective tissue, such as the weakening of tendons or ligaments following surgery or immobilization, and for the design of a new generation of mechanically tunable implants.

In the field of connective tissue regenerative medicine, the recent reviews by Huang et al. [82] and Liu et al. [83] provide comprehensive accounts which, although not explicitly mentioning MCT, implicitly include related topics and concepts; the first [82] offers a survey of the current available biomimetic scaffolds of natural or synthetic origin for tendon tissue engineering; the second [83] provides an updated overview of the available types of biomimetic natural biomaterials (BNBMs) with biological and physicochemical characteristics of native ECM. In the light of the latest advances in biomimetics, MCT-inspired synthetic substitutes (such as artificial tendons with differential tensility, and dynamically controllable ligaments) could be employed in the fields of: (1) personalized medicine, to temporarily or permanently replace, entirely or in part, damaged structures [82,83]; or (2) soft-robotics, to design and construct new-generation biomimetic actuators and multifunctional robotic materials [79,81,84,85,87] (see Section 4.5).

### 4.2. Technology-Derived Developments: Technology-Pull Processes

In this case, the starting point for knowledge transfer is a technical problem whose solution requires an appropriate technical product developed through an engineering-driven process [78]. Biological models are a source of inspiration, but the related biomimetic technical products do not necessarily appear to be morphologically similar to them and frequently do not display the same functions. Following these principles, a gradually increasing number of applied technology-pull processes inspired by echinoderm MCT have been proposed in biomedicine as well as in other areas, either in terms of advanced smart materials to be employed in diverse contexts, or in terms of soft actuators in soft robotics.

The pioneering example was developed by Capadona et al. [88], who assembled the first mechanically adaptive material inspired by sea-cucumber dermis, for the purpose of developing implantable intracortical microelectrodes. The importance of adaptability or stimulus responsiveness is critical for implanted materials, since a common cause of implant failure in vivo is the inability of engineered materials to adapt to their biological environment. The model of Capadona et al. [88] was both structurally and functionally biomimetic, since it reproduced the natural composite material structure by employing tunicate (sea-squirt)-derived cellulose ‘whiskers’, i.e., nanocrystals (t-CNCs), immersed in a polymeric matrix (ethylene oxide–epichlorohydrin 1:1 copolymer (EO-EPI) or polyvinyl acetate (PVAc)), and it also displayed adaptive chemoresponsive mechanical behavior (Figure 8).

In this model, mechanical changes (stiffening/destiffening) occurred in response to chemical stimuli through (1) inter-nanocrystal crosslinks (hydrogen bonds) that could be switched on or off, and (2) competitive nanocomposite–water interactions related to water uptake [91]. Exposure to water caused competitive hydrogen bond formation between water and cellulose nano-whiskers with a consequent stiffness reduction. It should be noted that, although all versions of this model were intrinsically biomimetic [88,89], they also utilized a biotechnological approach since they employed a bio-derived product as a structural material, i.e., cellulose nanocrystals extracted from tunicates or cotton. Advances in the field of intracortical implantation have included the development of mechanically compliant microelectrodes (again employing t-CNC nanocomposites) that are more compatible with the mechanical needs of the tissue, reduce long-term neurodegenerative and neuroinflammatory responses, and preserve both neuronal and glial integrity [90,92] (Figure 8). The significant role of new materials and electrode designs in this field was recognized by Faisal and Iacopi [93], who also considered the importance of research on their reliability and suitability in terms of biocompatibility, cytotoxicity, electrical and contact stability, and mechanical compatibility [94].

### 4.3. Advanced Materials

On the basis of their structural, chemical and mechanical characteristics as well as their easy production and commercial potential, bioinspired nanocomposites based on functionalized CNCs or CNFs (cellulose nanofibers) have attracted considerable scientific interest during the past 15 years. They mostly consist of CNCs or CNFs of diverse origins (Figure 9) immersed in a polymeric matrix and variously functionalized by additional components (reviewed by [95]). This has led to the successful employment of nanocellulose preparations in a wide range of sustainable material applications and particularly in the design of stimuli-responsive adaptive materials [17,79,96,97].

Thorough comparative studies of biological and synthetic models of ‘self-shaping materials’ have provided insights into the basic design principles and properties of a variety of responsive nanocomposite soft-materials that are employable in diverse applications (for reviews, see [98,99]). These have also elucidated the critical role of their hierarchical organization and molecular interactions in relation to multifunctional and N-dimensional properties [79].

**Figure 9 marinedrugs-22-00037-f009:**
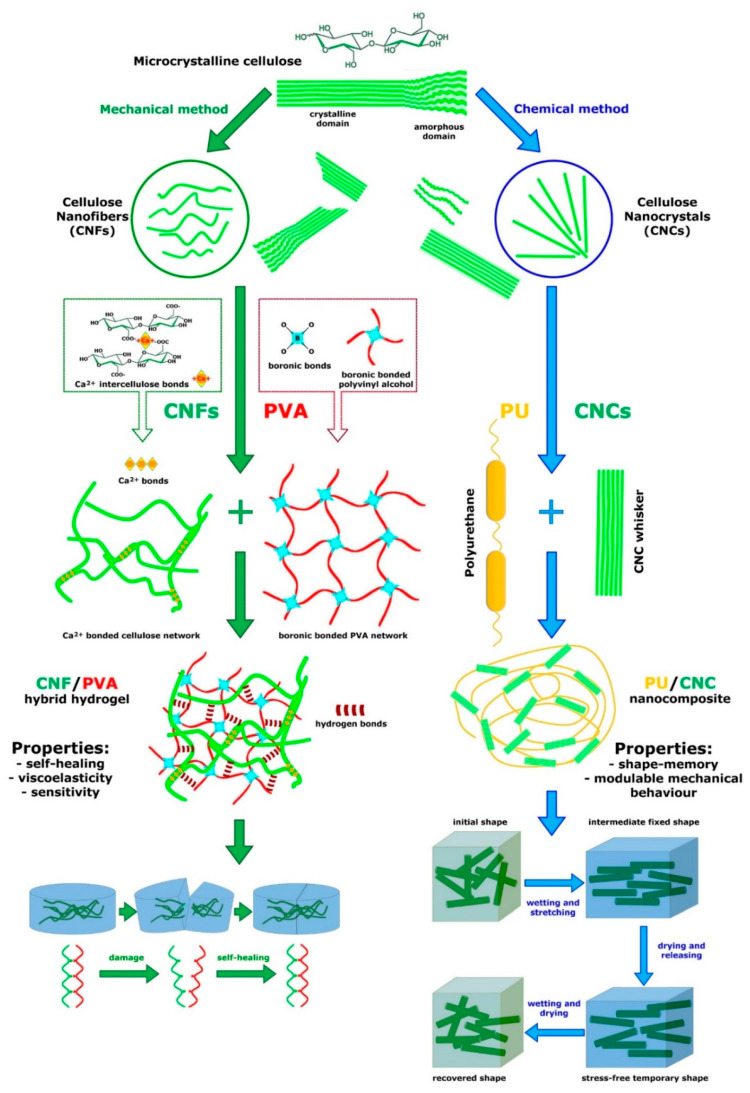
Examples of models of two bioinspired nanocomposites based on cellulose elements and polymeric matrix and characterized by different performances; both models can be considered analogous to the natural MCT model. On the left: basic structure of a hybrid hydrogel consisting of a network of mechanically isolated CNFs crosslinked with Ca^2+^ (green inset) and a network of PVA (polyvinyl alcohol) crosslinked by dynamic boronic bonds (red inset). The CNF/PVA hydrogel consists of a double network crosslinked by hydrogen bonds (redrawn and modified from [95]). Its main specific property is self-healing (explained below in main text) as schematically shown on the bottom left. On the right: basic structure of a shape-memory nanocomposite consisting of chemically isolated CNC whiskers (cellulose nanocrystals) immersed in PU (polyurethane) matrix (redrawn and modified from [100]). Orientation/alignment of CNC whiskers and crosslinks between them can be modulated according to wetting/drying and stretching conditions. The main property of the nanocomposite is shape-memory whose mechanism is shown on the bottom right (redrawn and modified from [91]).

A very common application of the sea-cucumber dermis model, inspired by its stiffening, destiffening and autolytic capabilities, has been adopted for the development of many hydrogels, i.e., three dimensional crosslinked polymeric networks swollen with water. Hydrogels, which exhibit very diverse chemical natures and composition, represent versatile and functional biomimetic systems that can respond to external and internal physical or chemical stimuli [79,96,101]. Their more frequent applications are in the biomedical and biotechnology areas (as targeted drug vehicles, tissue engineering matrices and bio-sensors) as well as in the agriculture and cosmetic industries. In particular, hybrid hydrogels are widely employed in biomedical fields because (1) they display characteristics of natural biomaterials, (2) they can integrate polymers with functional groups, and (3) their properties can be altered by the variation in polymers/molecules according to the requested requisites. Moreover, hydrogel systems are the most suitable ECM analogs for tissue engineering and cell cultures [102].

The long list of CNC/CNF-derived nanocomposite materials currently being utilized includes a variety of (multi)stimuli-responsive hybrid gels (Figure 9) with shape-memory, self-healing (Figure 9), adhesive and gluing properties [95,103,104,105]. Most of these cases were explicitly inspired by holothurian dermis and indirectly led to the development of other types of alternative mechanically adaptive biomimetic nanocomposites (derived, for example, from chitin or silk).

Due to their versatility, a new generation of ‘mechanically morphing’ structural materials with the capacity to sense, adapt and self-repair have been successfully developed [17,79,96,106]. In this scenario, an impressive number of stimuli-responsive nanocomposite systems with predictable and programmable behavior (‘smart’ or ‘intelligent’ materials) have been developed for applications where minimally invasive medical devices are required [82,98,99].

In their review of bioinspired polymer systems with stimuli-responsive mechanical properties, Montero de Espinosa et al. [107] discussed examples of sea-cucumber dermis-inspired materials that can reversibly alter their stiffness, shape, porosity, density or hardness upon remote stimulation. It is particularly relevant that the switching principles underpinning sea-cucumber dermis mutability have been used in the design of a number of shape-memory CNC-based nanocomposites (Figure 9) with diverse polymeric matrices (polyurethane, polyglycerol sebacate urethane, polyvinyl acetate, poly(butyl methacrylate), etc.), all capable of adopting one or more temporary shapes while ‘remembering’ their original shape [91,96,106]. CNC-based nanocomposites have been developed over the years, with starting properties modified by functionalizing CNCs with specific groups (amino or carboxylic acid groups, etc.), or adding other suitable components (carbon nanotubes, chitin-derived nanocrystals and others) [107]. Tian et al. [108] proposed polyurethane (PU) nanocomposite models reinforced by modified cellulose nanocrystals (CNC-UPy (2-ureido-4[1H]-pyrimidinone)), showing improved mechanical strength when compared with pure PU matrix, while maintaining elongation at break and toughness. Gorbunova et al. [109] designed nanocellulose-based thermo-, moisture-, and pH-sensitive polyurethane composites with shape memory properties and modulable mechanical performance.

Furthermore, the sea-cucumber-inspired approach has also been used to develop other types of mechanically adaptive nanocomposites. Xu et al. [110] proposed an elastomeric material (SCIM—Sea-Cucumber-Inspired Material), consisting of a hydrogen-bonded/Diels–Alder dynamic covalent dual-crosslinked network to which abundant Zn^2+^–imidazole crosslinks were also added. This system is based on reversible stiff/soft responses in dry–wet cycles. It shows remarkable changes in mechanical properties as well as recyclability and self-healing (the capacity to automatically repair structure and properties in response to damage), and has applications in the field of microelectrodes and actuators (see Section 4.5). Yuan et al. [111] designed a water-responsive hybrid polymer system with reversible and permanent covalent networks by crosslinking poly(propylene glycol) with boroxine and epoxy. The hybrid epoxy–boroxine displayed remarkable self-healing functionality and tunable mechanical properties and has been proposed for potential applications in transfer printing, nanoelectrode lithography and self-healable conductors/sensors.

Overall, on the basis of the recent successful development of a variety of biomimetic models of smart materials (which can sense and react to environmental stimuli), it is clear that nanocomposites can benefit from being basically designed with polymeric materials to bridge stiff components, in most cases, represented by nanocellulose elements. Heise et al. [95] underlined how nanocellulose-based nanocomposites display the widest range of applications (films, barrier layers, fibers, filtering membranes, structural colors, gels, aerogels and foams, viscosity modifiers, etc.), although still presenting challenges in the design of their detailed nano-structure and assembly in relation to their multi-functional properties. The recent review by Solhi et al. [97] gives a comprehensive up-to-date account of the use of nanocellulose in a broad range of sustainable material applications, focusing in particular on nanocellulose−water interactions and their implications.

Starting from initial applications where water movement was employed to regulate hydrogen-bonding interactions in the nanocomposite framework, many studies have involved models that respond to other, more specific physicochemical environmental stimuli, such as temperature or pH changes, UV irradiation, and light [17,107,112]. In particular, very different types of stimuli-responsive hybrid materials, all inspired by sea-cucumber dermis, have been recently developed, as shown in the following examples.

In terms of light-responsive mechano-adaptive materials, molecular switches that can modify the shape and stiffness of soft materials in response to light stimuli have been investigated. Through the incorporation of artificial molecular switches into heterogeneous anisotropic soft matter (represented by liquid crystal polymer networks including free liquid crystals), Lancia et al. [113] designed innovative responsive ‘actuating’ materials that displayed complex and fast mechanical adaptability and deformability, including both light-induced stiffening and softening, and nonlinear responses to stress, comparable to that of MCT and muscles. Later, Jiao et al. [114,115] demonstrated how mechanical properties of switchable polymeric materials can adapt to light irradiation [114] and to low-voltage direct current (DC) [115]. The authors designed adaptive, highly-reinforced nanocomposites (cellulose nanofibrils combined with water-soluble copolymers incorporating thermo-reversible bonds) and, through the employment of laser irradiation [114] or low DC [115], induced, respectively, photothermal or electrothermal energy transfer cascades, causing reversible modulation of mechanical properties.

As far as magnetic field responses are concerned, Kobayashi et al. [116] proposed a magnetic-responsive smart material inspired by holothurian MCT consisting of magnetic elastomer polypropylene glycols and tetrapod-shaped zinc oxide elements. They demonstrated relevant changes in viscoelastic properties by testing weak magnetic fields on samples in which magnetic-responsive components were about 20% of the bulk material volume.

These and other types of smart materials that are able to respond to low-intensity stimuli derived from the external environment (humidity, temperature, light) have been widely employed in biomedicine, particularly in the fields of tissue engineering and regenerative (TER) medicine, biocompatible substrates, implantable devices and robotic materials (see Section 4.4 and Section 4.5). The study by Yang [102] presented a novel multifunctional, stimuli-responsive hydrogel based on carboxymethyl-chitosan (CM) and acrylamide (AM). This hydrogel displayed biomimetic functions such as reversible stiffness changes, shape-memory abilities and self-healing properties, and appeared to be applicable in the biomedical area in the form of motion sensor devices, ‘on-demand’ switches in electric circuits, and 3D printing inks, as illustrated by the examples proposed by the author.

In addition to its wide biological applications in biomedicine and other areas, MCT behavior has been a source of inspiration for smart materials with non-biological industrial uses. Myronidis et al. [19] successfully developed a novel protective smart layer to be employed on the surface of carbon fiber-reinforced polymer (CFRP) laminates. The proposed material (Shear Stiffening Gel—SSG) consists mainly of a polyborosiloxane-based (PBS) polymer and displays unique characteristics, being able to modify the way energy is distributed during impact events, due to a dynamic transition between viscous and rubbery phases.

### 4.4. Biocompatible Substrates

There has recently been a rapid development of soft biomaterials in the biomedical area in parallel with advanced technologies. For example, smart biomaterials characterized by good biocompatibility and sensitive responses to environmental changes, such as thermal self-healing and shape-memory responses, have attracted particular interest [117].

Due to their high biocompatibility, structural similarity with collagenous tissues, and functional performance (mechanical and swelling properties), nanocellulose-based hydrogels have been used in the fields of (1) tissue engineering, (2) injectable substrates, and (3) sensors and actuators (see Section 4.5). In the first application, nanocellulose-based hydrogels are widely employed as an extracellular matrix scaffold for growth and regeneration of a variety of tissues such as bone, cartilage, heart, blood vessel, nerve, and liver (for a review, see [118]). Their swollen three-dimensional network structure is quite similar to that of connective tissue for the diffusion of signals and molecules (nutrients, growth factors, etc.) and for mediating cell functions [118]. As injectable substrates, nanocellulose-based hydrogels are particularly advantageous because of their ability to be delivered to a target site via a minimally invasive route (for reviews, see [79,119,120]).

As far as MCT-inspired biomimetic systems are concerned, Gao et al. [120] synthesized sea-cucumber-inspired supramolecular polymer (SP) hydrogels obtained by direct photoinitiated aqueous polymerization of N-acryloyl-2-glycine monomers. Due to their reversible multiple non-covalent bonds, the hydrogels exhibited tunable mechanical properties, repairability, and reusability. In vitro cytotoxicity tests and preliminary subcutaneous implantation indicated that supramolecular polymer hydrogels can be biocompatible and autolytic in vivo, with the potential to be used as temporary devices for intestinal drug delivery or as an injectable filling to assist the suturing of small vessels.

In their review on smart polymers, Huang et al. [117] provided a comprehensive account of specific characteristics (self-healing, shape-memory) and responsiveness (to thermal, chemical, electric, magnetic, light, pH and humidity stimuli) of three main types of smart polymeric materials. The authors also considered future progress and challenges in the field, explaining how the tunable properties and environmental responses of smart polymeric materials can provide an opportunity to design personalized biomedical products.

Table 1 summarizes the main features of the MCT-analogous materials discussed above in Section 4.3 and Section 4.4.

### 4.5. Soft Actuators and Robotic Tools

The recent review by Li et al. [121] provides a comprehensive account of different types of successfully developed biomimetic actuators, particularly hydrogels, directly inspired by natural mechanisms and with attractive properties, such as adaptability, stimuli responsiveness, controllability, flexibility/toughness and reconfigurability. The basic principles of stimuli responsiveness in polymeric materials are reviewed and illustrated with both natural models (including echinoderm MCT) and bioinspired hydrogel actuators, focusing in particular on the mechanisms of hydrogel mechanical toughening and its key-role in applications [121].

With regard to soft robotics, soft actuators are essential components that affect robot performance in terms of actuation patterns, movements and resilience [18]. A starfish-like soft robot (bionic model) with flexible rays and multi-gait locomotion was developed using Shape-Memory Alloy (SMA) actuators and 3D printing technology [122]. More recently, other soft robotic actuators inspired by starfish and sea-urchins were proposed and successfully tested in different environments. These include fluidically actuated arms, variable friction appendages, and tube foot-inspired suction-based actuators (reviewed by Bell [123] and Picardi et al. [124]).

Taking inspiration from many other animal models, a new generation of self-operating soft actuators has been recently developed, which are based on different actuation mechanisms driven by water, electricity, heat, magnetism and chemical reactions. Of particular interest are water-driven self-operating actuators undergoing motion through a swelling/deswelling process. Choi et al. [18] developed strong, dynamic, poly(N-isopropylacrylamide) hydrogels inspired by MCT, whose elastic modulus could be altered significantly via hydration/dehydration with a shape-memory effect. Based on the hydrogel stiffness changes obtained by variations in crosslinker concentration, a programmable, powerful and ultra-fast water-driven self-operating soft actuator was developed (SITuator), which is able to modulate the actuation force and speed and to preserve its original shape over multiple cycles of actuations in different environments. According to Choi et al. [18], a variety of robotic applications of SITuator could be developed to perform a wide range of diverse tasks.

In the field of soft robotics, other models have been explicitly inspired by MCT. One example, proposed by Liu et al. [125] is an impact-protective SPM material characterized by unique impact-hardening and reversible stiffness-switching based on the dynamic aggregation of nanoscale elements in soft polymeric networks, modulated by H-bonding. SPM appears as an analog of sea-cucumber dermis in terms of both structure (i.e., hierarchical nanostructures and transient polymeric networks) and properties (i.e., viscoelasticity and smart stiffness–softness switching). Liu et al. [125] illustrated its potential for protective applications with examples of impact-resistant and puncture-resistant robotic devices.

Li et al. [126] used an integrated biomimetic design to fabricate another successful sea-cucumber MCT analog using sidechain polypseudorotaxanes with tunable nano- to macro-scale properties. On the basis of their physicochemical characteristics, the size of the polypseudorotaxane crystalline domain and the hydrogel crosslinking density can be adjusted, thereby improving the mechanical adaptability of the material for employment in 3D printing. After 3D printing and photo-crosslinking, the hydrogels demonstrated stiffness changes, exhibiting large tensile strain and wide elastic-to-plastic variations.

Soft materials and actuators can be successfully applied in situations where traditional rigid robots and actuators cannot be employed [81,87]. Due to their adaptability, soft actuators can be employed in complex and dynamic environments and in physical interactions with fragile objects or living organisms. In particular, soft actuators and robotic devices are being increasingly developed and applied in the field of rehabilitation/assistance for improving a range of medical treatments. In comparison with rigid devices, soft actuators and robotic tools can offer significant advantages, including improved mobility, safer human–machine interaction, easier fabrication, and greater resilience. Although not mentioning possible MCT-inspired applications, the reviews by Li et al. [87] and Pan et al. [81] offer comprehensive accounts of the potential of new materials for new-generation soft actuators characterized by physical intelligence and advanced properties (adaptability, multimodal locomotion, self-healing and multi-responsiveness). These authors also explain how performance and multi-functionality can be implemented using programmable bioinspired soft materials [87] and they highlight important real-world applications of soft actuators [81].

Taking inspiration from stimuli-responsive movements of animals and plants, Wang et al. [85] designed a new class of bionic actuators consisting entirely of biocompatible and biodegradable materials and providing a good solution for in vivo biomedical soft robotics and biomimetic devices. Stimuli-responsive genetically engineered silk–elastin-like proteins (SELPs) and CNFs were combined in an actuator system able to respond to physical and chemical stimuli with programmable and reversible deformations.

Regarding the development of new soft robots, Nasseri et al. [127] showed how stimuli-responsive hydrogels can be employed as versatile soft actuators by introducing microstructural anisotropy and shape-change programmability in their design. The proposed nanocomposite hydrogels, largely composed of zwitterionic monomers and asymmetric CNCs, displayed anisotropic swelling and mechanical properties as well as self-healing properties and cytocompatibility.

The review by Tang et al. [84] focuses on mobility and responsive systems and offers an overview addressed specifically to soft actuator motion. Due to their functional analogy to soft biological tissues (specifically muscles, but also other tissues), soft actuators are obviously integral components of soft robots: they can be activated by fluid, thermal, electric, magnetic, light, humidity and chemical stimuli, giving rise to bending, linear, torsional, spiral and composite movements actuated by a variety of specific mechanisms and employable in a wide range of application fields. In this scenario, the future broad employment of MCT-inspired soft actuators in different applied fields appears to be a very realistic prospect (Figure 10).

Although the basic mechanisms underpinning holothurian dermis mechanical adaptability have already been mimicked directly or indirectly in the field of soft robotics, the performance of the artificial models is still far from optimal and is characterized by generally slower and less complex responses in comparison with natural models. Future efforts must, therefore, be aimed at designing systems that, emulating natural MCT, involve the faster and more effective intervention of intrinsic switching agents released in response to external stimuli [107].

## 5. Future Prospects

We conclude that it is likely that MCT will continue to be exploited as a source of inspiration for the development of new biomimetic materials. As is the case with many natural processes, the molecular mechanisms underpinning the mechanical adaptability of MCT are complex, which makes them difficult to reproduce in artificial optimized models. Furthermore, although the biological aspects of MCT have been investigated for over 50 years, there are still major gaps in our knowledge. To achieve progress in the field of MCT-inspired biomimetics, the immediate aims should be: (1) to acquire a progressively deeper understanding of the structural–functional relationships in the biological models at all hierarchical levels [79,107,121] and to apply these relationships to engineering materials [86]; (2) to develop an interdisciplinary approach that utilizes the expertise of biologists, biomimeticists, materials scientists and software engineers; and (3), by taking advantage of advanced computer programs, to achieve more cost- and time-efficient design processes through the control of structural and functional features on different hierarchical levels via integrated computational modeling and processing.

## Figures and Tables

**Figure 1 marinedrugs-22-00037-f001:**
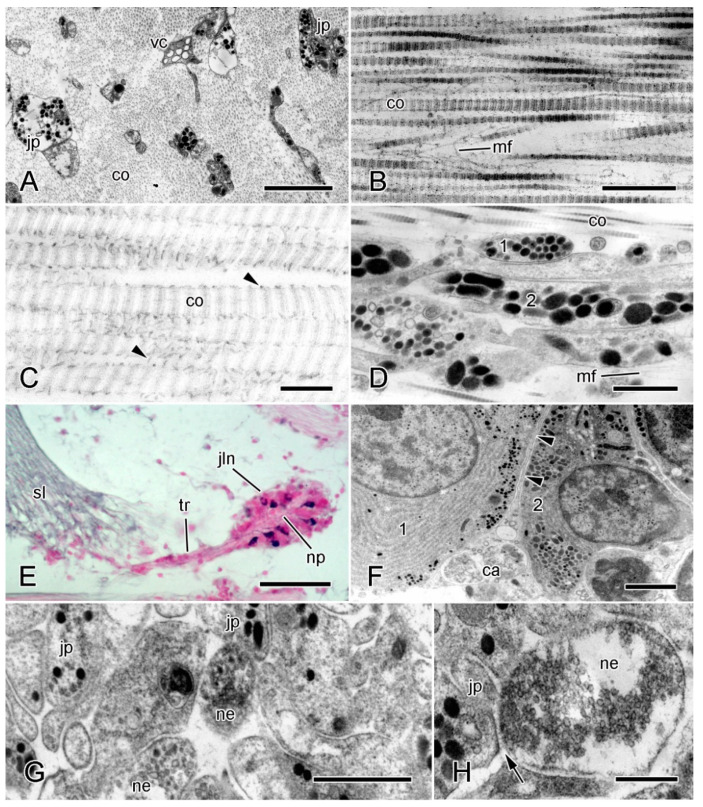
Structural organization of MCT. (**A**,**B**) Transmission electron micrographs (TEMs) of intervertebral ligament of brittlestar *Ophiocomina nigra*. (**A**) Transverse section. Scalebar = 2 µm. (**B**) Longitudinal section. Scalebar = 0.5 µm. (**C**) TEM (longitudinal section) of brachial ligament of featherstar *Antedon bifida*, stained with cupromeronic blue, which labels interfibrillar proteoglycans (arrowheads). Scalebar = 0.2 µm. (**D**–**H**) Juxtaligamental cells associated with the spine ligament of *O. nigra*. (**D**) TEM (longitudinal section) of spine ligament showing two types of LDCV-containing juxtaligamental cell process (1,2). Scalebar = 1 µm. (**E**) Light micrograph (transverse section) of juxtaligamental node and tract of juxtaligamental cell processes extending into spine ligament; stained with chrome hematoxylin and phloxine. Scalebar = 50 µm. (**F**–**H**) TEMs of juxtaligamental node. (**F**) Edge of node showing two types of juxtaligamental cell (1,2) and outer capsule forming intercellular partition (arrowheads). Scalebar = 2 µm. (**G**) Neuropil-like region. Scalebar = 1 µm. (**H**) Chemical synapse (arrow) between neuronal and juxtaligamental cell processes in neuropil-like region. Scalebar = 0.5 µm. ca, capsule; co, collagen fibrils; jln, juxtaligamental node; jp, juxtaligamental cell process; mf, microfibril; ne, neuronal cell process; np, neuropil-like region; sl, spine ligament; tr, tract of juxtaligamental cell processes; vc, heterogeneous vacuole-containing cell. ((**B**) modified from [21] with permission of John Wiley & Sons-Books; permission conveyed through Copyright Clearance Center, Inc.). (**C**) Courtesy of U. Welsch. (**D**,**F**,**G**) Modified from [22], https://doi.org/10.1371/journal.pone.0167533 (accessed on 2 May 2023), under the terms of the CC BY 4.0 license, https://creativecommons.org/licenses/by/4.0, accessed on 2 May 2023).

**Figure 2 marinedrugs-22-00037-f002:**
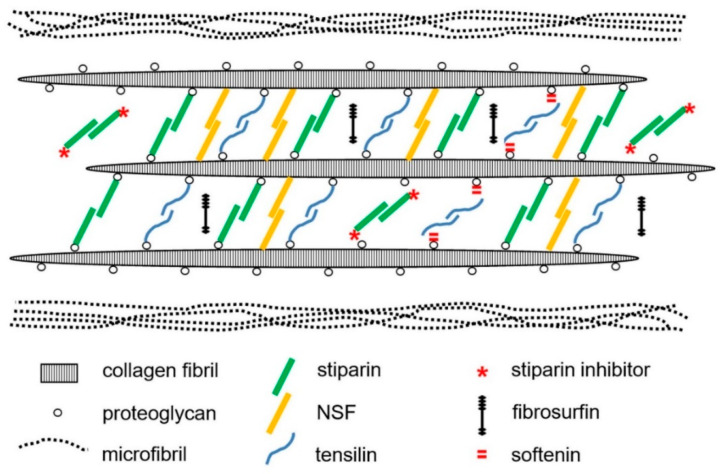
Model of MCT molecular organization based primarily on sea-cucumber dermis and including some of the factors known to influence its mechanical properties. For simplicity, the model assumes that stiparin, novel stiffening factor and tensilin form dimers that act as interfibrillar crossbridges. Fibrosurfin has been detected in sea-urchin MCT but its function and microstructural disposition are unknown. (Used with permission of Royal Society of Chemistry, from reference [20]; permission conveyed through Copyright Clearance Center, Inc.).

**Figure 3 marinedrugs-22-00037-f003:**
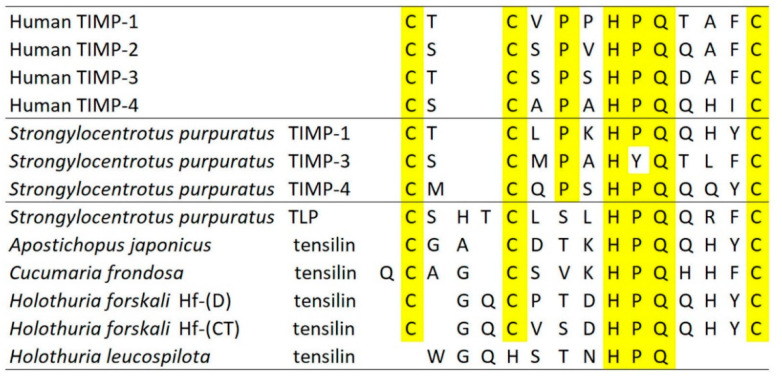
Partial N-terminal domain amino acid sequences of human and *Strongylocentrotus purpuratus* TIMPs and of echinoderm tensilins, including sea-urchin tensilin-like protein (TLP). Identical amino acids are highlighted. Sources of data: references [22,23,24]. (Modified from [20] with permission of Royal Society of Chemistry; permission conveyed through Copyright Clearance Center, Inc.).

**Figure 4 marinedrugs-22-00037-f004:**
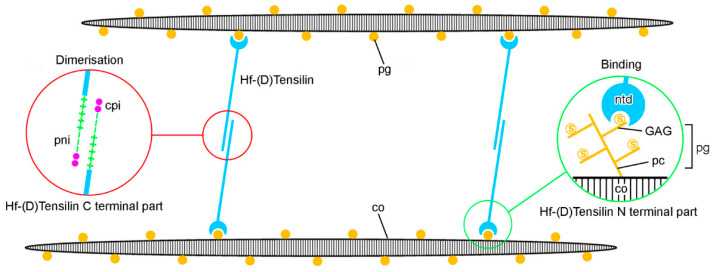
Model of molecular interactions between Hf-(D)Tensilin and collagen fibrils in dermis of sea-cucumber *Holothuria forskali*. Crosslinking of collagen fibrils (co) depends on two types of interaction: (1) dimerization or oligomerization of tensilin molecules via positively and negatively charged amino acid interactions (pni) and non-covalent cation-π interactions (cpi) at the C-terminal part of tensilin (red circles), and (2) binding of the NTR-TIMP-like domain (ntd) at the N-terminal part of tensilin with sulfates (s) on glycosaminoglycan (GAG) sidechains of proteoglycans (pg) attached via their protein core (pc) to the surface of collagen fibrils (green circles). (Redrawn and modified from reference [45]).

**Figure 5 marinedrugs-22-00037-f005:**
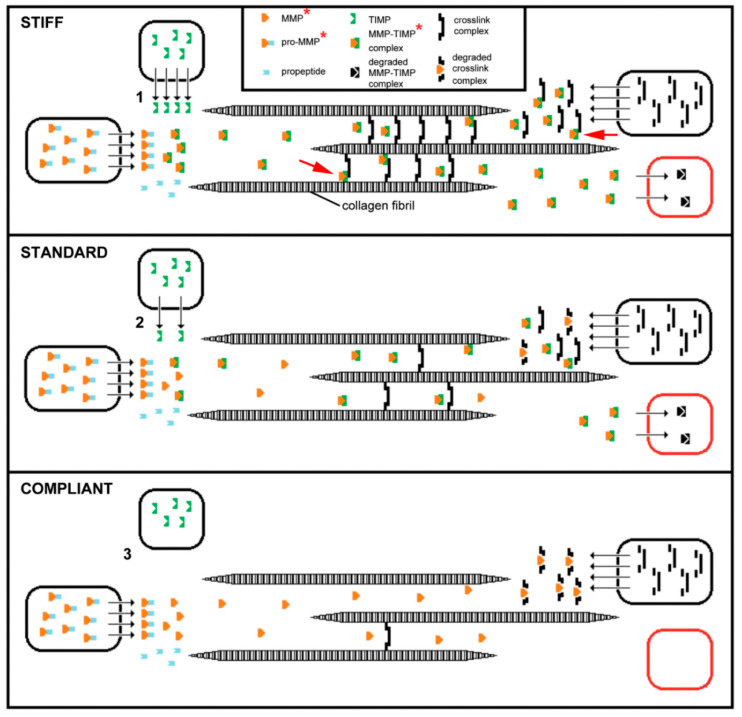
Hypothetical model of the involvement of MMPs in MCT mutability, inferred from data on the compass depressor ligaments (CDLs) of the sea-urchin *Paracentrotus lividus*. The synthetic MMP inhibitor galardin increases the stiffness of CDLs in all three mechanical states, which suggests that in all three states, there is (a) ongoing MMP activity that has the potential to degrade components already incorporated into existing crosslink complexes and components that have been secreted but not yet incorporated, and (b) ongoing synthesis and release of new crosslink components. MMPs are synthesized and secreted as inactive pro-enzymes, then activated extracellularly by proteolytic removal of the pro-peptide domain. It is envisaged that crosslink components are synthesized and secreted separately, then assembled extracellularly to form functional complexes. The black boxes represent cells, although the three processes do not necessarily occur in different cell-types. The red box represents the process by which MMP-TIMP complexes are removed and degraded. The model assumes that activated MMPs and new crosslink components reach the extracellular environment at a constant rate. It is hypothesized that interfibrillar cohesion is regulated only through changes in the rate at which an endogenous MMP inhibitor (which is assumed to be a TIMP-like molecule) is released into the extracellular environment. In the stiff state, there are high levels of TIMP secretion (1), MMP inhibition, and crosslinking. In the standard state there are intermediate levels of TIMP secretion (2), MMP inhibition, and crosslinking. In the compliant state there are low levels of TIMP secretion (3), MMP inhibition, and crosslinking. Also represented is the possibility that an endogenous inhibitor could function as a component of the crosslink complex (red arrows) and thus have a dual function (which may apply to TIMP-like tensilin). The model also assumes that the production of MMP-TIMP complexes exceeds the rate of removal and degradation of MMP-TIMP complexes, which would account for the finding that there is a positive correlation between the degree of CDL stiffness and total gelatinolytic activity. The components marked with a red asterisk contribute to the gelatinolytic activity of CDLs as quantified by gelatin zymography. (From reference [65], https://doi.org/10.1371/journal.pone.0049016 (accessed on 2 May 2023), under the terms of the CC BY 4.0 license, https://creativecommons.org/licenses/by/4.0, accessed on 2 May 2023).

**Figure 6 marinedrugs-22-00037-f006:**
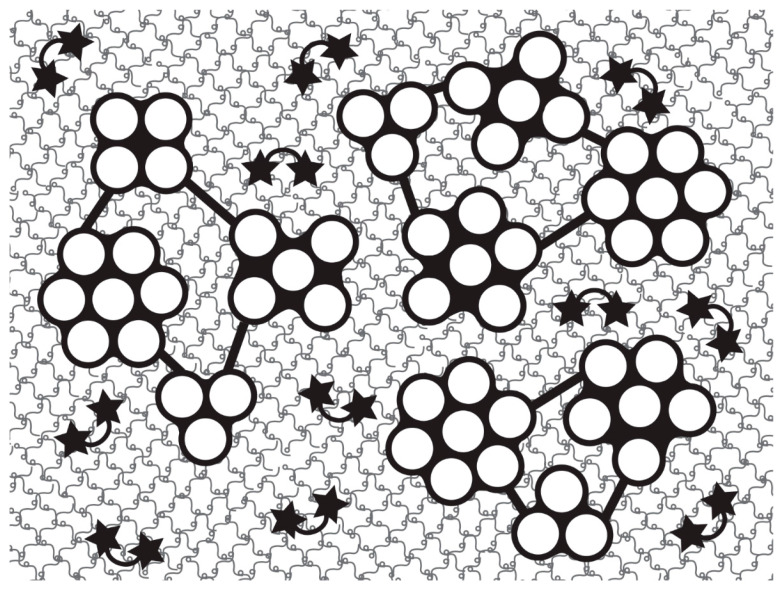
Model of holothurian dermis. The dermis is modeled as a material with three structural levels. Level 1, collagen fibrils: each fibril is an aggregation of subfibrils (white circles) embedded in a matrix (black filling between subfibrils); the stiffness of the matrix is controlled through tensilin and softenin. Level 2, fibril bundles (fibers): each bundle consists of fibrils linked by crossbridges (black bars) and embedded in a matrix whose stiffness is controlled by the number of crossbridges. Level 3, dermis: the bundles are embedded in a hydrogel of proteoglycans (represented as tadpole-shaped molecules) whose stiffness is controlled by bonds between proteoglycans, bond formation being associated with water exudation; the arcs with a star at each end denote bonds between proteoglycans. (From reference [71], https://doi.org/10.1371/journal.pone.0155673 (accessed on 2 May 2023), under the terms of the CC BY 4.0 license, https://creativecommons.org/licenses/by/4.0, accessed on 2 May 2023).

**Figure 7 marinedrugs-22-00037-f007:**
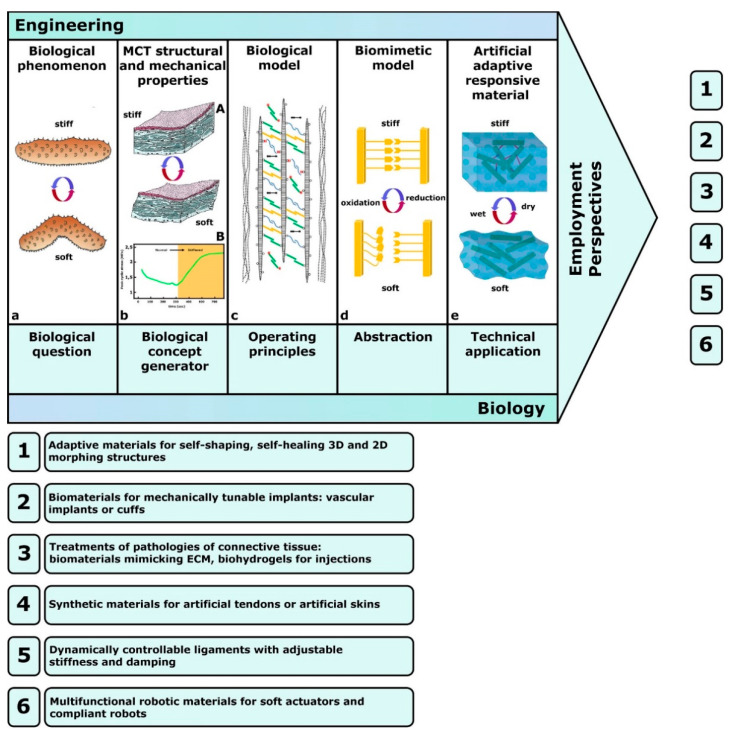
The bottom-up biomimetic approach (biology-push process) inspired by MCT according to the model proposed by Trotter et al. [16] (arrangement freely inspired by [77]). (**a**) The biological model: sea-cucumber dermis. (**b**) The basic structural (A) and functional (B) features of the mutability phenomenon (B: redrawn and modified from [72]). (**c**) Natural MCT model and its main components: the functional principle is based on dynamic interactions between collagen fibrils and ECM proteoglycans and glycoproteins (see Figure 2 of this review). (**d**) Basic scheme of a simple MCT-inspired biomimetic model as proposed by Trotter, consisting of a synthetic fibrous composite with dynamic interactions between the components. In the stiff condition, crosslinks are formed by the interaction of catechol and phenylboronic acid. In the soft condition, crosslinks are reversed by the oxidation of catechol to orthoquinone (modified from Trotter, unpublished). (**e**) Technical applications of this operating principle in a fibrous nanocomposite with dynamically controlled stiffness (redrawn and modified from [17]). (1–6) Employment perspectives of MCT-inspired models.

**Figure 8 marinedrugs-22-00037-f008:**
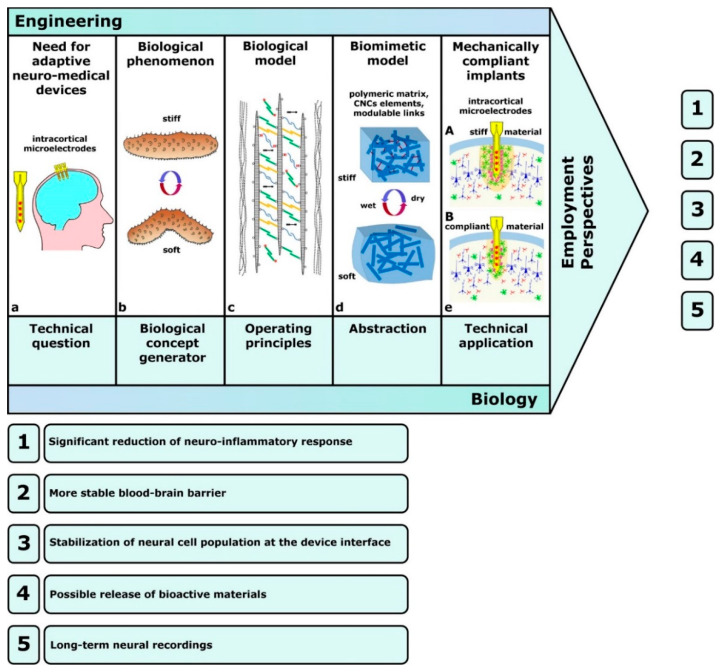
The top-down biomimetic approach (technology-pull process) inspired by MCT according to the model proposed by Capadona et al. [88] (arrangement freely inspired by [78]). (**a**) Example of a neuro-medical device requiring a biocompatible adaptive substratum: implantable intracortical microelectrode. (**b**) The biological model: sea-cucumber dermis. (**c**) Natural MCT model and its main components as in Figure 7c. (**d**) Basic MCT-inspired biomimetic model: polymeric soft matrix reinforced by CNC rigid elements interconnected by modulable links (redrawn and modified from [89]). (**e**) Employment of MCT-inspired material for mechanically-compliant implants: comparison of neural tissue response to standard stiff (A) and compliant (B) intracortical microelectrodes (redrawn and modified from [90]). (1–5) Advantages of mechanically-compliant microelectrodes in intracortical implants.

**Figure 10 marinedrugs-22-00037-f010:**
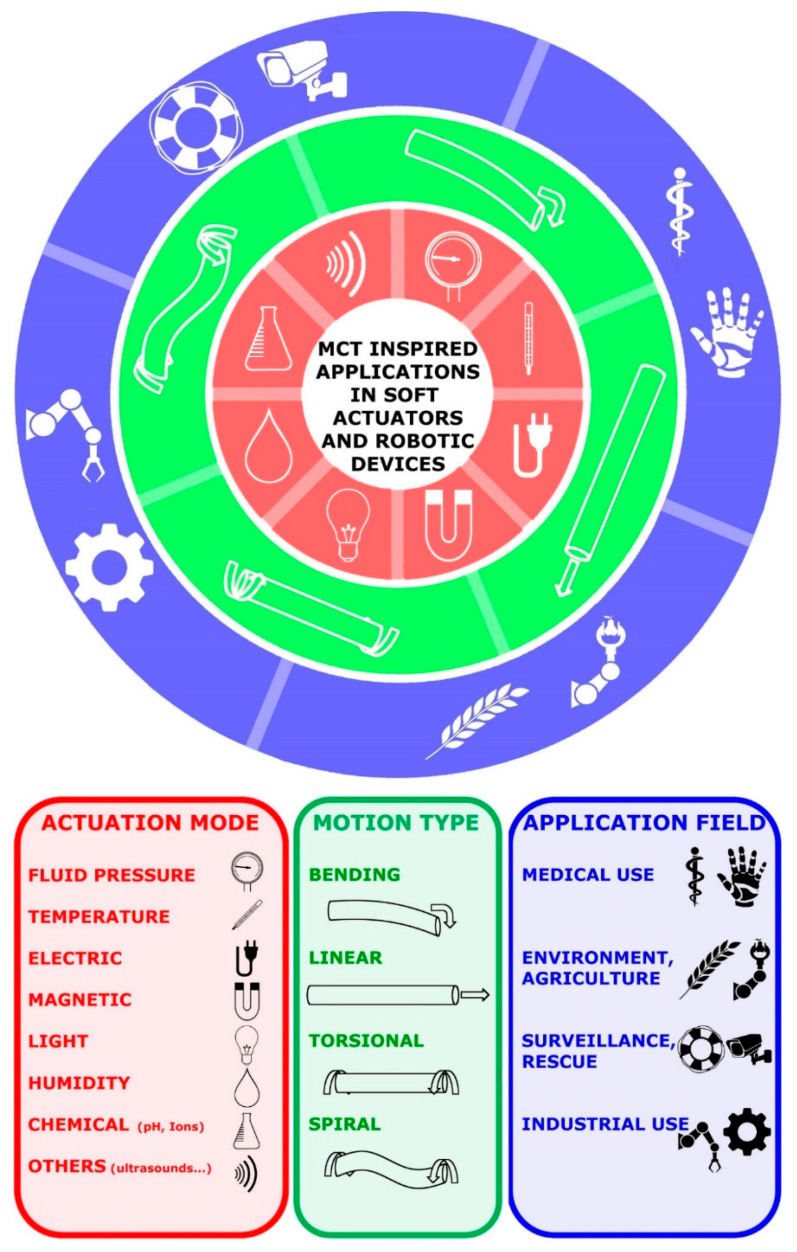
Actuation modes, motion types and application fields of presumptive MCT-inspired soft actuators and robotic devices (redrawn and modified from [84]).

**Table 1 marinedrugs-22-00037-t001:** Main features of MCT-inspired advanced materials and substrates.

Composition	Characteristics	Employability	Main Applications
**Nanocellulose components:** CNCs/CNFs (functionalized or chemically modified or not); CNC/CNF derivatives **Polymeric matrix:** PVA, UE, PU, and many others **Possible additional components (bioderived or artificial):** carbon nanotubes, etc.	**Advantages:** ease of production; sustainability; high commercial potential **Structural** **characteristics:** hierarchical organization; self-shaping, self-assembling; molecular interactions **Functional properties:** shape-memory; self-healing; gluing and adhesiveness; foldability; multi-stimuli responsiveness; modulable mechanical behavior; actuation abilities **Specific biological functionalities:** biocompatibility; biodegradability; low cytotoxicity; low invasivity; injectability; reusability	Biomedical area Tissue engineering Regenerative medicine Cosmetics Biotechnological area Agriculture Laminate industry	Multifunctional and diverse hybrid hydrogels; 2D and 3D scaffolding Cell culture adaptable substrata Injectable biocompatible fluids and substrata Tendon, ligament and skin repair/regeneration Ophthalmic applications Fingerprint detection; 3D printing inks; electric circuits Self-healable conductors/sensors Multifunctional robotic materials Magnetic elastomers Protective layers for non-biological materials

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
