# Peer review of "Mutable Collagenous Tissue: A Concept Generator for Biomimetic Materials and Devices"

_marinedrugs, 2024, doi:10.3390/md22010037_

Round 1

Reviewer 1 Report

Comments and Suggestions for Authors

Candia Carnevali and co-authors present a concise up-to-date overview of the echinoderm mutable collagenous tissue biology. They then delve into how that foundational understanding inspires the development of innovative bioinspired materials in various applications ranging from regenerative medicine to robotics. The manuscript is very well-written and constitutes an informative read for both experts and newcomers in the field.

I have only two minor points for improvement.

First. In the Fig. 1 legend, all abbreviations should be arranged in alphabetical order.

Second. Lines 115-125. The surveys of echinoderm genomes and transcriptomes did not reveal any “unusual” components in the echinoderm ECM. Can that be due to the fact that those surveys compare the echinoderm genes to databases of previously annotated genes in other animals? That way, unknown genes that are potentially unique to echinoderms might not be detected. This, in turn, would mean that there might be more still undiscovered molecular components in the echinoderm MCT that remain to be discovered.

Author Response

RESPONSE TO REVIEWER 1

We thank reviewer 1 for their comments.

Point 1. “First. In the Fig. 1 legend, all abbreviations should be arranged in alphabetical order.”

Response. All abbreviations are now in alphabetical order.

Point 2. “Second. Lines 115-125. The surveys of echinoderm genomes and transcriptomes did not reveal any “unusual” components in the echinoderm ECM. Can that be due to the fact that those surveys compare the echinoderm genes to databases of previously annotated genes in other animals? That way, unknown genes that are potentially unique to echinoderms might not be detected. This, in turn, would mean that there might be more still undiscovered molecular components in the echinoderm MCT that remain to be discovered.”

Response. We agree, this is possible. To cover this point, we have added a sentence to emphasise that comparative -omics information on echinoderm ECM is very limited. Please see first sentence of 3rd paragraph of Section 2.1.

Reviewer 2 Report

Comments and Suggestions for Authors

Evaluation of the manuscript marinedrugs-2770214

The aim of this review is to give a detailed description of the molecular mechanisms underlying the stiffening processes of mutable collagenous tissue in echinoderms, based on the scientific literature of the last 20 years. Moreover, it provides an overview of the main applications in which the concept of mutable collagen has been an inspiration for the creation of smart biomaterials in different fields.

Overall, the review is very detailed and comprehensive, encompassing all the most recent bibliographical references in the sector. It demonstrates a profound knowledge of the topic by the authors involved. In the description of the stiffening mechanism of collagenous tissues, the review thoroughly explores all the molecular and cellular effectors involved. In my opinion, the review is suitable for publication; however, I have listed some corrections and suggestions below:

-       Line 187. Here the Authors stated that “the inextractability of collagen fibrils from normal adult vertebrate collagenous tissues”. However, to avoid confusion in the less experienced reader, I suggest changing it specifying first that collagen is easily isolated in molecular form as tropo-collagen from adult vertebrate tissues (Ref) but is inextractable in its form organized into fibers.

-       Line 660. Why the authors define “nanocomposites” nanocellulose-based hydrogels if they are made of only one type of polymer?

-       A graphical representation of the main cells involved in mutable collagen is strongly suggested.

-       The authors present the alignment of the tensilins with the TIMPs in Figure 3, but there are no comments in the text regarding the potential key role of specific amino acids in the tensilins. I recommend including this information.

-       Finally, to enhance readability, I suggest inserting tables to support paragraphs 4.3 and 4.4. These tables should summarize the main applications mentioned in the text along with their compositions, characteristics, and corresponding bibliographic references.

Author Response

RESPONSE TO REVIEWER 2

We thank reviewer 2 for their comments.

Point 1. “Line 187. Here the Authors stated that “the inextractability of collagen fibrils from normal adult vertebrate collagenous tissues”. However, to avoid confusion in the less experienced reader, I suggest changing it specifying first that collagen is easily isolated in molecular form as tropo-collagen from adult vertebrate tissues (Ref) but is inextractable in its form organized into fibers.”

Response. This has been corrected as advised.

Point 2. “Line 660. Why the authors define “nanocomposites” nanocellulose-based hydrogels if they are made of only one type of polymer?”

Response. “Nanocomposites” has been removed.

Point 3. “A graphical representation of the main cells involved in mutable collagen is strongly suggested.”

Response. We regret that we do not know what exactly reviewer 2 means by “graphical representation”. The main cells involved in MCT are juxtaligamental cells. Their microanatomical relations and ultrastructure are already fully illustrated in Figure 1.

Point 4. “The authors present the alignment of the tensilins with the TIMPs in Figure 3, but there are no comments in the text regarding the potential key role of specific amino acids in the tensilins. I recommend including this information.”

Response. We do not know if reviewer 2 is referring only to amino acids in the partial N-terminal sequences shown in Figure 3 or not. This problem, and the absence of literature references directing us to relevant information, makes us unable to comply with this recommendation. We also believe that the amount of detail on tensilin function that is already provided in paragraph 2 of Section 3 and in the caption of Figure 4 is sufficient for the intended readership.

Point 5. “Finally, to enhance readability, I suggest inserting tables to support paragraphs 4.3 and 4.4. These tables should summarize the main applications mentioned in the text along with their compositions, characteristics, and corresponding bibliographic references.”

Response. We have added Table 1. Please note that we have not included references, as this would have complicated the table and would have been an unnecessary duplication of information already in the main text.

Reviewer 3 Report

Comments and Suggestions for Authors

In the Manuscript titled “Mutable Collagenous Tissue: A Concept Generator for Biomimetic Materials and Devices”, the Authors describe the basic physiology and molecular architecture of mutable collagenous tissue (MCT) and review applications in the fields of regenerative medicine and biosensors based on the conceptual framework of MCT, applications that do not per se utilize biological materials extracted from echinoderm MCT.

Overall comments: For most readers, MCT will be a novel concept developed in a well-resourced review. The Authors should be commended for embarking on such a novel idea and developing it with extensive references. However, and even though the Authors have dedicated a large effort to clarifying the novelty of MCT-inspired thinking for medical and other types of applications, the Manuscript requires significant editorial modifications to add robustness to its exposition.

Major issues: Extensive editing needed to improve coherence, eliminate ambiguity, and increase readability.

Minor concerns: Others, please see attached file.

Comments on the Quality of English Language

The quality of English is good

Author Response

RESPONSE TO REVIEWER 3

We thank reviewer 3 for their comments.

Point 1. “Major issues: Extensive editing needed to improve coherence, eliminate ambiguity, and increase readability.”

Response. We have complied with most of reviewer 3’s suggestions, as explained below in our response to point 2.

Point 2. “Minor concerns: Others, please see attached file.”

Response. We have complied with most of the suggestions made by reviewer 3 in their annotated version of our MS (i.e., marinedrugs-2770214-reviewer 2.pdf). We show how we have responded in the attached file (marinedrugs-2770214-reviewer 2 RESPONSE.pdf). Please note that the latter shows only our responses to reviewer 3’s comments; we have not changed the text in this version.

We have complied with reviewer 3’s suggestions, except where these obscure or remove an important point, or result in an incorrect meaning.

Round 2

Reviewer 3 Report

Comments and Suggestions for Authors

The Authors have successfully addressed the comments and suggested changes, improved readability of the document, and the Manuscript is now ready for the next step in the publication process.